# Genome-wide study of linkage disequilibrium, population structure, and inbreeding in Iranian indigenous sheep breeds

S. Barani[1], A. Nejati-Javaremi[1], M. H. Moradi[2]*, M. Moradi-Sharbabak[1], M. Gholizadeh[3], H. Esfandyari[4]

1 Department of Animal Science, University College of Agriculture and Natural Resources, University of Tehran, Karaj, Iran, 2 Department of Animal Science, Faculty of Agriculture and Natural Resources, Arak University, Arak, Iran, 3 Department of Animal Science, Sari Agricultural Sciences and Natural Resources University, Sari, Mazandaran, Iran, 4 TYR, Hamar, Norway

* Hoseinmoradi@ut.ac.ir, H-Moradi@Araku.ac.ir

**Data Availability Statement:** All relevant data are included within the paper.

## Abstract

Knowledge of linkage disequilibrium (LD), genetic structure and genetic diversity are some key parameters to study the breeding history of indigenous small ruminants. In this study, the OvineSNP50 Bead Chip array was used to estimate and compare LD, genetic diversity, effective population size ($N_e$) and genomic inbreeding in 186 individuals, from three Iranian indigenous sheep breeds consisting of Baluchi ($n = 96$), Lori-Bakhtiari ($n = 47$) and Zel ($n = 47$). The results of principal component analysis (PCA) revealed that all animals were allocated to the groups that they sampled and the admixture analysis revealed that the structure within the populations is best explained when separated into three groups (K = 3). The average $r^2$ values estimated between adjacent single nucleotide polymorphisms (SNPs) at distances up to 10Kb, were 0.388±0.324, 0.353±0.311, and 0.333±0.309 for Baluchi, Lori-Bakhtiari and Zel, respectively. Estimation of genetic diversity and effective population size ($N_e$) showed that the Zel breed had the highest heterozygosity and $N_e$, whereas the lowest value was found in Baluchi breed. Estimation of genomic inbreeding using $F_{ROH}$ (based on the long stretches of consecutive homozygous genotypes) showed the highest inbreeding coefficient in Baluchi and the lowest in Zel breed that could be due to higher pressure of artificial selection on Baluchi breed. The results of genomic inbreeding and $N_e$ showed an increase in sharing haplotypes in Baluchi, leading to the enlargement of LD and the consequences of linkage disequilibrium and haplotype blocks confirmed this point. Also, the persistence of the LD phase between Zel and Lori-Bakhtiari was highest indicating that these two breeds would be combined in a multi-breed training population in genomic selection studies.

**Funding:** This study was funded by Animal Science Research Institute of Iran, Mobarakandish Institute and AgResearch, New Zealand, Project number: PRJ-2016/11547. The funders had no role in study design, data collection and analysis, decision to publish, or preparation of the manuscript.

**Competing interests:** The authors have declared that no competing interests exist.

# Introduction

Archaeological and genetic evidence suggest that domestication of sheep occurred approximately 9000 ago (BC) in a region in the west of the Zagros Mountains of Iran-Iraq [1]. Furthermore, according to the FAO in 2020, Iran is sixth in the world for the live sheep with around 47 million sheep (https://www.fao.org/faostat/en/#data/QCL). Iran has been blessed with several different climates that produce various sheep breeds. Therefore, different climates have accelerated diversity in the genetic structure of Iranian sheep. In a worldwide climate-changing scenario, keeping animals adapted to harsh environmental conditions becomes increasingly important. In this sense, local sheep breeds constitute an important genetic resource due to their rusticity and adaptability to various agroecological environments. The Iranian sheep breeds used in this study are among the most common and main indigenous sheep breeds reared in a variety parts of Iran, that being collected from or near the center of domestication. The Baluchi is a fat-tailed medium size breed that is well adapted to the warm and dry environmental conditions and is accounting over 29% of Iranian sheep population [2]. The Lori-Bakhtiari sheep is one of the most widespread native breeds in the southwestern of the Zagros mountains, and has the fattest tail of all Iranian sheep breeds those adapted to cold and mountainous regions of western Iran. Whereas, the Zel is the only thin tail Iranian breed adapted to the lush green and wet conditions of northern slopes of the Elburz mountain range near the Caspian Sea. This breed is also known as the Aryan breed since the historical evidence shows that the Aryans, who were living in these areas, attempted to domesticate these animals [3].

The knowledge of genetic structure is crucial for genomic prediction within and among populations, genome wide association and local sheep breeds conservation. Rapid progress in the genomic selection (GS) will facilitate the opportunity to implement GS in small ruminant [4]. The basic assumption in GS is that the marker haplotypes should be in linkage disequilibrium (LD) with the quantitative trait loci (QTLs) located between the markers, and the useful threshold for LD in GS and association studies should be higher than 0.3 [5]. LD is defined as the non-random relationship between two loci within a population [6]. Currently, the genetic relationship between multiple breeds is identified by analyzing LD, consistency of gamete phase, and haplotype block structure between different breeds. Thus, when the markers and QTLs have similar LD phase between the breeds, we can use informative markers of both breeds in construction of a multi-breed training population [7].

Amid the important challenges of genome-wide studies on small ruminants is the restriction on artificial insemination that leads to striving for pedigree information based on multiple-sire natural mating groups of rams and ewes in a natural mating system that is not reliable. Then, the knowledge of inbreeding, genetic diversity, and the effective population size ($N_e$) of livestock populations is crucial for the success of breeding programs. The development of genome wide information in small ruminants has allowed for the measuring of genomic inbreeding and diversity by identifying runs of homozygosity (ROH) and heterozygosity. In livestock genetics, ROH regions, consisting of continuous homozygous loci assumed to originate from the same ancestor, are commonly used for inbreeding detection [8]. Furthermore, The estimation of historical effective population size (Ne) is a widely spread method of modelling the evolution of genetic diversity of populations and it is very useful for designing conservation strategies of indigenous breeds [9].

Therefore, considering that the uneven distribution of LD along the genome has an important effect on genomic prediction, in such a way that genetic variance in regions of high LD is overestimated for causal variants and is underestimated in regions of low LD [10], we investigated the genetic structure of Iranian indigenous sheep breeds by LD as the first objective and, the second objective was to estimate observed and expected heterozygosity, ROH, and $N_e$.

## Material and methods

### Ethics statement

All methods and animal care and handling procedures were allowed and approved by the University of Tehran Animal Care and Use Committee (No. 2016/11547). All efforts were carried out in accordance with relevant regulations to minimize any discomfort during blood collection. The authors also complied with the ARRIVE (Animal Research: Reporting of In Vivo Experiments) guidelines.

### DNA samples and SNP genotyping

Three sources of genomic data, including Zel (n = 47), Lori-Bakhtiari (n = 47), and Baluchi (n = 96), were used in the current study. The dataset of Zel and Lori-Bakhtiari has been previously described in detail by Moradi et al. [11] and Baluchi sheep by Golizadeh et al. [12]. Animal sampling for the Zel breed was performed in the northern region, for the Lori-Bakhtiari breed in the western part of Iran close to the Zagros Mountains, and the Baluchi breed in the Abbas-Abad sheep breeding station in north-eastern Iran. DNA was extracted from all animal blood samples using salting out methods [11] and then, genotyping was performed using the Illumina Ovine SNP50 Bead Chip.

### Quality control (QC) and genetic diversity analyses

Quality controls of the genomic data was performed using PLINK v1.09 software [13] as follows; first animals with more than 10% missing genotypes were removed. Then, the SNPs with minor allele frequency (MAF) <0.05 and call rates <95% over all samples, and SNPs deviating strongly from Hardy-Weinberg equilibrium within breed (P-value<$10^{-6}$) were excluded. The Hardy-Weinberg equilibrium was tested as genotype errors, as it is most likely that technical problems explain this result [14]. To obtain a significant level in this test, the Bonferroni correction ($\beta = \alpha/n$) was used to address the problem of multiple comparisons. The number of tests was taken to be the number of SNPs ($n$ = 50,000) giving $\beta = 10^{-6}$, which corresponds to $\alpha$ = 0.05 experiment-wise error [15].

For the remaining SNPs after mentioned QC, the SNP locations were mapped for ovine genome assembly by OAR_V4.0 using information from the Sheep HapMap dataset (http://www.sheephapmap.org); and finally, the SNP markers whose genomic location was unknown or located on the X chromosome were removed from the total markers. Expected heterozygosity ($H_e$) and observed heterozygosity ($H_o$) were calculated for each SNP which passed the quality control following the methods suggested by Al-Mamun [16], and then averaged over all SNPs.

### Principal component analysis (PCA) and population analyses

Principal component analysis (PCA) was performed using the prcomp function in the R version 4.0.3 package (http://cran.r-project.org), which considers the total variance in the data and alters the original variables into a smaller set of linear compounds. After that for more insight and confirmation, population clustering and admixture analysis were determined using the program ADMIXTURE 1.23 [17]. ADMIXTURE is a "hill-climbing" optimization algorithm as a pre-compiled binary executable based on maximum-likelihood that estimates a $F_{ST}$ value based on the inferred allele frequencies between each of the ancestral populations [18]. We ran Admixture for 10 to 20 iterations increasing K (An input value for belief of the number of ancestral populations [17]) from 2 to 4. Cross validation (CV) error estimation for each K was performed to determine the optimal number of clusters.

## Measures of average LD and persistence of phase

Haplotype reconstruction and phasing of the genotypes by chromosome were carried out using BEAGLE 3.3.1 [19]. A haplotype is a physical grouping of genomic variants from multiple genetic loci on the same chromosome that are inherited as a unit that can encompass two or more SNP alleles [20]. The linkage disequilibrium between adjacent SNPs was measured by the squared correlation coefficient ($r^2$) [21]. The $r^2$ was computed based on the following equation by Haploview v4.2 software [22]:

$$r^2 = \frac{(FreqAB*Freqab - FreqAb*FreqaB)^2}{(FreqA*Freqa*FreqB*Freqb)} \qquad \text{Eq 1}$$

Where $freqA$, $freqa$, $freqB$ and $freqb$ are the frequencies of alleles A, a, B, b, respectively, and $freqAB$, $freqab$, $freqAb$ and $freqaB$ are the frequencies of haplotypes AB, ab, Ab and aB, respectively. Then, based on estimated $r^2$ values, sample size correction was performed by the following equation [23]:

$$r^2_{corrected} = \frac{r^2_{computed} - \frac{1}{n}}{1 - \frac{1}{n}} \qquad \text{Eq 2}$$

Where $n$ is the number of haplotypes in the sample. Averaged $r^2$ was calculated for chromosomes of each breed between pairwise SNPs in different distance categories (0.01<, 0.01–0.02, 0.02–0.04, 0.04–0.06, 0.06–0.08, 0.08–0.1, 0.1–0.2, 0.2–0.5, 0.5–1, 1–2, 2–5, 5–10 and 10–20 (Mb)).

To estimate persistence of LD phase between two breeds, only segregating SNPs in both breeds were included in the analysis. Persistence of LD phase was estimated for intervals of 100 kb following Badke et al. [24] as:

$$R_{kk'} = \frac{\sum_{(ij)\in p} \left( r_{ij(k)} - \bar{r}_{(k)} \right) \left( r_{ij(k')} - \bar{r}_{(k')} \right)}{S_{(k)}S_{(k')}} \qquad \text{Eq 3}$$

Where $R_{kk'}$ is the correlation of phase between $r_{ij(k)}$ in population k and $r_{ij(k)}$ in population k', $S_{(k)}$ and $S_{(k)}$ are the standard deviation $r_{ij(k)}$ and $r_{ij(k)}$, and $\bar{r}_{(k)}$ / $\bar{r}_{(k')}$ are the average $r_{ij}$ across all SNP $i$ and $j$ within interval $p$ for population $k$ and $k'$, respectively.

**Haplotype blocks.** The haplotypes were phased using BEAGLE v3.3.1 software [19], and then the haplotype blocks were determined for each chromosome using Haploview v4.2 software [22] based on the method suggested by Gabriel et al. [25]. A pair of markers was defined to be in strong LD if the one-sided 95% confidence bound of $D'$ was higher than 0.98 and if the lower bound was over 0.7.

## Inbreeding coefficients and effective population size

Inbreeding coefficients based on genotype data for each breed were calculated by GCTA software [26]. The three estimates of inbreeding coefficient ($F$) calculated by this program consist of the $F_{GRM}$ calculated based on the variance of the additive genotype, the $F_{HOM}$ estimated based on the excess of homozygotes, and the $F_{UNI}$ calculated based on the correlation between uniting gametes. Inbreeding coefficient for each breed was measured as the average of inbreeding coefficients of all individuals for that breed. Runs of homozygosity (ROH) was calculated using PLINK v9.1 software [13] with adjusted parameters (—*homozyg-density 1000*,—*homozyg-kb 10*,—*homozyg-window-het 1*, *and*—*homozyg-window-snp 20*). The minimum number of SNPs needed to constitute an ROH (l) was estimated using the

method proposed by Lencz et al. [27],

$$l = \frac{log_e \frac{\alpha}{n_s.n_i}}{log_e(1 - het)}$$

Eq 4

Where $n_s$ is the number of genotyped SNPs per individual, $n_i$ is the number of individuals, $\alpha$ is the percentage of false-positive ROH (0.05), and *het* is the average of SNP heterozygosity across all SNPs. Finally, the measurement of inbreeding based on ROH ($F_{ROH}$) was calculated by the following equation [27]:

$$F_{ROH} = \frac{L_{ROH}}{L_{AUTO}}$$

Eq 5

Where $L_{ROH}$ is the sum of ROH lengths and $L_{AUTO}$ is the total length of autosomes.

Effective population size ($N_e$) is a key population genetic parameter for calculating genetic diversity and can be used to estimate the inbreeding coefficients. We estimated effective population sizes from LD using SNeP software [28]. The software SNeP allows the estimation of $N_e$ trends across generations using SNP data that corrects for sample size, phasing, and recombination rate.

$$N_{T(t)} = (4f(c_t))^{-1}\left(E\left[r_{adj}^2 \mid c_t\right]^{-1} - \alpha\right)$$

Eq 6

Where $N_T$ is the effective population size $t$ generations ago, calculated as $t = (2f(c_t))- 1$ [29], $c_t$ is the recombination rate defined for a specific physical distance between markers, $r_{adj}^2$ is the LD value adjusted for sample size, and $\alpha = (1)$ a correction for the occurrence of mutations [30].

According to the relationship between $r^2$ and $N_e$, effective population size can be calculated from LD data for each autosomal chromosome at distance bins of <0.01, 0.01–0.02, 0.02–0.05,

**Table 1. Summary of quality control (QC) steps on genotyping data of autosomal SNPs.**

| Breed | Zel | Lori-Bakhtiari | Baluchi |
|---|---|---|---|
| Total animal before QC | 47 | 47 | 96 |
| Total SNP before QC | 51,103 | 51,103 | 51,103 |
| Excluding animals with call rate <10% | 2 | 2 | 8 |
| Excluding SNPs≤0.05 MAF | 4330 | 4742 | 6359 |
| Excluding SNPs ≤0.95 Call Rate | 1760 | 1893 | 187 |
| Excluding SNPs out of HWE ≤0.000001 | 11 | 6 | 5 |
| Excluding SNPs with unknown position | 746 | 746 | 746 |
| Excluding SNPs on sex chromosomes | 1165 | 1176 | 1166 |
| Remaining Animals after QC | 45 | 45 | 88 |
| Remaining SNPs after QC | 43091 | 42540 | 42640 |

MAF: minor allele frequency, HWE: Hardy-Weinberg equilibrium

0.05–0.1, 0.1–0.2, 0.2–0.5, 0.5–1, 1–2, 2–5, 5–10 and 10–20 *cM* by the following equations [31]:

$$E(r^2) \approx \frac{1}{(4cN_e + 1)} \qquad \text{Eq 7}$$

$$N_e = \left( \frac{1}{E(r^2)} - 1 \right) \cdot (1/4c) \qquad \text{Eq 8}$$

Where $N_e$ is the effective population size, $r^2$ is a measure of LD among SNP alleles per chromosome, and *c* is the recombination rate in Morgan units regarding the average distance between two markers, we assumed 1 Mb = 1 cM and the generation of $N_e$ or *T* is equal to 1/2*c* [29].

## Results and discussion

### Quality control (QC) and a summary of statistics obtained for the SNPs passed QC

Out of 190 animals from three sheep breeds, 178 animals passed the quality control. Table 1 shows the results of quality control for each breed.

The average distance between SNPs on autosomal chromosomes after filtering was 57.92, 58.78 and 58 Kb in Zel, Lori-Bakhtiari and Baluchi breeds, respectively. The average marker distances in current study for Zel and Lori-Bakhtiari were different from 60Kb that reported by Moradi et al. [15]. This may be due to the use of the Ovine Genome Assembly V4.0, instead of v1.1 that have been used by Moradi et al. [15], and also excluding the sexual SNPs in the current study.

A summary of the distribution of the remaining SNPs after quality check per each chromosome and the average $r^2$ on each chromosome is indicated in Table 2.

Chromosome 21 had the largest distance between adjacent SNPs for all breeds (Table 2). The maximum and the minimum number of SNPs over all genotyped animals were observed on chromosome 1 and 24, respectively, for all breeds. The minor allele frequency (MAF) is important genetic diversity parameter and effective on population structure [32, 33]. It influences the $r^2$ value, and $r^2$ decreases significantly with the difference in MAF between loci [34]. O'Brien et al. [35] proposed MAF>0.05 as the optimal threshold for QC and estimated unbiased LD.

In our study, the average of MAF for the SNPs before quality control (QC) was 0.27, 0.27, and 0.25, in Zel, Lori-Bakhtiari and Baluchi, respectively, while these values were changed to 0.29, 0.29, and 0.28 after QC check with MAF>0.05. McRae et al. [36] reported the average of MAF between 0.24 up to 0.26 using same genotyping array in New Zealand sheep. The distribution of MAFs is affected by the long-term demography of the population it represents [37]. It seems New Zealand sheep have been under more selection intensity during last years, and are thus more inbred [38], resulting in slightly lower MAFs. The average of MAF for Iranian sheep breeds was almost similar to what reported by García-Gámez et al. [39] in Spanish Churra sheep breed (the average of MAF = 0.288). The distribution of MAF per each chromosome was almost uniform in different breeds. The proportion of SNPs with the MAF higher than 0.3 was 48.3%, 51.36%, and 47.8% in Baluchi, Lori-Bakhtiari, and Zel, respectively.

The average $r^2$ on each chromosome showed that OAR 24 and 25 in Baluchi, OAR 9 and 21 in Lori-Bakhtiari and OAR 23 and 24 in Zel have higher LD value than other chromosomes, while in Australian sheep breeds OAR 10, 22 and 23 had the highest $r^2$ and haplotype blocks [16]. Also, Liu el al. [40] demonstrated OAR 24, 25, 18 and 10 had the highest average $r^2$ values

**Table 2. A summary of statistics obtained for the SNPs passed quality control (QC) in different chromosomes of three Iranian sheep breeds used in this study.**

| | Number of SNP | | | Average of MAF ± SD | | | Average distance between SNPs (kb) | | | r2 ± SD | | |
|---|---|---|---|---|---|---|---|---|---|---|---|---|
| Chr | Zel | Lori-Bakhtiari | Baluchi | Zel | Lori-Bakhtiari | Baluchi | Zel | Lori-Bakhtiari | Baluchi | Zel | Lori-Bakhtiari | Baluchi |
| 1 | 4727 | 4690 | 4731 | 0.289±0.128 | 0.293±0.124 | 0.295±0.125 | 57.27 | 58.73 | 58.21 | 0.025±0.034 | 0.029±0.038 | 0.027±0.044 |
| 2 | 4634 | 4562 | 4558 | 0.283±0.128 | 0.290±0.124 | 0.292±0.126 | 53.66 | 54.5 | 45.55 | 0.025±0.035 | 0.029±0.039 | 0.028±0.046 |
| 3 | 4067 | 4036 | 4006 | 0.288±0.128 | 0.292±0.127 | 0.295±0.126 | 55.08 | 55.51 | 55.92 | 0.025±0.034 | 0.030±0.04 | 0.026±0.042 |
| 4 | 2251 | 2225 | 2234 | 0.287±0.128 | 0.287±0.125 | 0.288±0.126 | 52.87 | 53.49 | 53.28 | 0.025±0.034 | 0.029±0.04 | 0.028±0.046 |
| 5 | 1958 | 1937 | 1916 | 0.286±0.127 | 0.289±0.125 | 0.289±0.125 | 54.56 | 55.16 | 55.76 | 0.025±0.034 | 0.028±0.038 | 0.029±0.047 |
| 6 | 2144 | 2125 | 2071 | 0.286±0.129 | 0.289±0.127 | 0.292±0.126 | 54.16 | 54.65 | 55.22 | 0.025±0.034 | 0.028±0.038 | 0.030±0.05 |
| 7 | 1844 | 1812 | 1831 | 0.286±0.129 | 0.292±0.126 | 0.290±0.125 | 53.97 | 54.93 | 55.36 | 0.025±0.034 | 0.029±0.039 | 0.027±0.044 |
| 8 | 1735 | 1725 | 1710 | 0.287±0.129 | 0.300±0.127 | 0.294±0.124 | 52.13 | 52.49 | 52.97 | 0.024±0.034 | 0.029±0.039 | 0.029±0.048 |
| 9 | 1772 | 1757 | 1716 | 0.294±0.126 | 0.293±0.125 | 0.297±0.128 | 53.13 | 53.58 | 54.87 | 0.025±0.034 | 0.031±0.041 | 0.030±0.049 |
| 10 | 1502 | 1483 | 1431 | 0.272±0.133 | 0.295±0.128 | 0.292±0.126 | 56.43 | 57.15 | 59.23 | 0.025±0.035 | 0.029±0.041 | 0.030±0.057 |
| 11 | 971 | 943 | 976 | 0.283±0.128 | 0.286±0.128 | 0.284±0.128 | 63.12 | 64.99 | 62.70 | 0.025±0.034 | 0.028±0.038 | 0.025±0.042 |
| 12 | 1438 | 1401 | 1404 | 0.288±0.127 | 0.287±0.127 | 0.293±0.128 | 54.76 | 56.20 | 56.09 | 0.025±0.034 | 0.029±0.040 | 0.028±0.047 |
| 13 | 1420 | 1379 | 1390 | 0.282±0.128 | 0.297±0.125 | 0.289±0.130 | 58.36 | 60.09 | 59.66 | 0.025±0.035 | 0.029±0.040 | 0.027±0.045 |
| 14 | 966 | 944 | 950 | 0.283±0.127 | 0.293±0.127 | 0.293±0.127 | 64.71 | 66.21 | 65.70 | 0.025±0.034 | 0.028±0.039 | 0.026±0.045 |
| 15 | 1391 | 1379 | 1369 | 0.283±0.130 | 0.294±0.126 | 0.293±0.126 | 57.56 | 58.06 | 58.48 | 0.025±0.034 | 0.028±0.038 | 0.031±0.051 |
| 16 | 1281 | 1276 | 1275 | 0.287±0.129 | 0.290±0.124 | 0.288±0.126 | 55.63 | 55.95 | 56.00 | 0.025±0.033 | 0.030±0.040 | 0.027±0.044 |
| 17 | 1156 | 1125 | 1190 | 0.283±0.128 | 0.292±0.123 | 0.294±0.126 | 62.35 | 64.04 | 60.55 | 0.025±0.035 | 0.030±0.040 | 0.026±0.045 |
| 18 | 1181 | 1166 | 1202 | 0.285±0.131 | 0.289±0.127 | 0.292±0.124 | 57.11 | 57.85 | 56.05 | 0.025±0.034 | 0.029±0.040 | 0.027±0.043 |
| 19 | 1008 | 979 | 1014 | 0.283±0.128 | 0.293±0.127 | 0.293±0.125 | 59.63 | 61.36 | 59.27 | 0.024±0.034 | 0.029±0.040 | 0.027±0.045 |
| 20 | 905 | 899 | 912 | 0.286±0.130 | 0.289±0.126 | 0.292±0.126 | 55.87 | 56.19 | 55.33 | 0.025±0.034 | 0.028±0.038 | 0.027±0.042 |
| 21 | 709 | 705 | 728 | 0.283±0.129 | 0.302±0.122 | 0.299±0.122 | 70.24 | 70.63 | 68.40 | 0.025±0.035 | 0.031±0.041 | 0.029±0.047 |
| 22 | 916 | 909 | 909 | 0.286±0.128 | 0.287±0.125 | 0.287±0.126 | 55.09 | 55.62 | 55.51 | 0.025±0.035 | 0.030±0.041 | 0.027±0.046 |
| 23 | 916 | 893 | 923 | 0.292±0.127 | 0.296±0.128 | 0.288±0.126 | 67.63 | 69.42 | 67.16 | 0.026±0.035 | 0.030±0.040 | 0.031±0.051 |
| 24 | 615 | 614 | 613 | 0.299±0.126 | 0.302±0.125 | 0.297±0.124 | 68.18 | 68.29 | 68.40 | 0.026±0.035 | 0.030±0.041 | 0.033±0.055 |
| 25 | 839 | 844 | 831 | 0.270±0.127 | 0.290±0.125 | 0.293±0.125 | 53.94 | 53.58 | 54.42 | 0.025±0.034 | 0.030±0.041 | 0.033±0.054 |
| 26 | 749 | 733 | 755 | 0.288±0.129 | 0.296±0.128 | 0.292±0.125 | 58.53 | 59.80 | 58.06 | 0.025±0.034 | 0.029±0.039 | 0.030±0.048 |

for a distance of 0–10 kb which is consistent with our results. Mateescu & Thonney [41] identified some regions on seven chromosomes 1, 3, 12, 17, 19, 20 and 24 in Dorset×East Frisian sheep, that associate with a seasonal reproduction trait. The difference in extent of LD on chromosomes in different studies could probably be due to the fact that LD means correlation between alleles, not physical association of loci, so factors other than physical distance on chromosomes such as mutation, genetic drift, epistasis and amalgamating two populations with different allele frequencies cause disequilibrium between unlinked markers [42].

## Population structure and genetic diversity

We performed principal component analysis (PCA) to identify how animals allocated to their true population in this study. The results clustered three distinct populations according to geographical origins and type breed (Fig 1). The first and second PCs ($PC_1$ and $PC_2$) accounted for 7.26 and 1.85% of the total variation, respectively. We found that $PC_2$ separated out thintail (Zel) and fat-tail (Lori-Bakhtiari and Baluchi) sheep breeds from each other, while fat tail sheep breeds were separated for $PC_1$ (Fig 1).

In the Baluchi population, some outliers were observed farther than the main population. The Lori-Bakhtiari and Zel populations have been subjected to smallholder sheep farms with

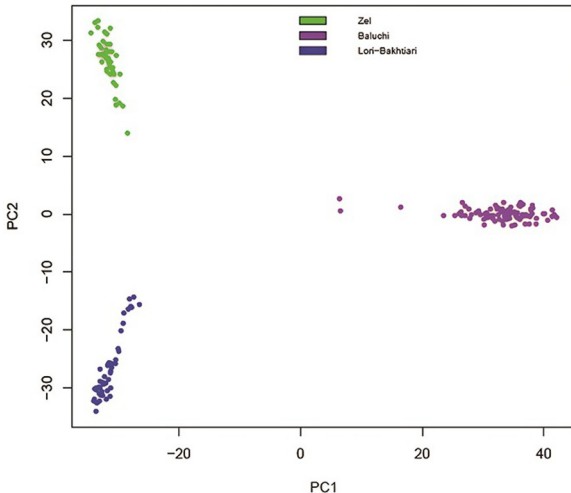

**Fig 1. Animals clustered based on principal components analysis using individual genotypes.** Green, purple and blue colors are showing the individuals of Zel, Baluchi and Lori-Bakhtiari sheep breeds in this Fig, respectively.

an extensive selection, incomplete pedigree, and uncontrolled mating, although the Baluchi population has been isolated in Abbas-Abad sheep breeding station. However, genetic links among the Lori- Bakhtiari and Zel populations are most presumably, arising by the absence of full pedigree information, co-ancestry, or gene flow. Until recently, the movement of livestock without animal identification had not been prohibited; therefore, gene flow could be possible due to animal migration by nomadic tribes.

Population structure was analyzed by considering different *K* numbers (2–4) based on autosomal chromosomes. The structure analysis with *K* = 2 clustered the Lori-Bakhtiari and Zel populations into the same group (Fig 2). It may be compatible with this belief that domestication of sheep occurred in the Zagros mountains, then outspread in other regions [1]. In other

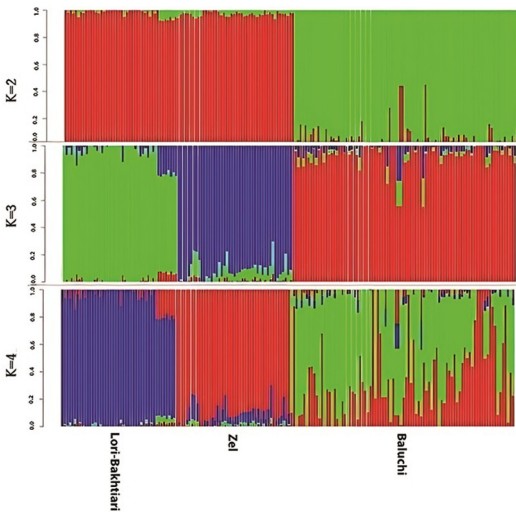

**Fig 2. Population structure of 178 individuals for three Iranian indigenous sheep breeds using genome-wide SNP data.** Each thin vertical line represents one individual and each color shows one inferred ancestral population.

words, this could be due to the migration of individuals between these two populations and possible common ancestry. These results represent the genetic closeness between Zel and Lori-Bakhtiari and confirm the findings based the PCA. Setting $K = 3$ clustered all populations into distinct clusters. When the $K$ value was 4, the structure analyses indicated introgression of the Zel population with the Baluchi population; however cross-validation error had the lowest value. It seems the influence of the Great Silk Road on the gene pool of local sheep to be plausible [43]. Cross-validation error for $K = 2$, 3 and 4 was 0.559, 0.554, and 0.553, respectively.

Various methods have been suggested to evaluate the genetic diversity, although the $H_o$ and $H_e$ are the most widely used to measure genetic diversity in a population [44]. The average of $H_e \pm SD$, calculated based on autosomal chromosomes, was $0.375 \pm 0.117$ for Baluchi and $0.382 \pm 0.113$ for both Zel and Lori-Bakhtiari and, the averaged $H_o$ for Baluchi, Zel, and Lori-Bakhtiari was 0.382, 0.383, and 0.388, respectively. It should be noted that the low heterozygosity observed in Balochi may be due to ascertainment bias, since the samples of this breed was collected from AbbasAbad station located in north-eastern Iran. Different animal breeding programs have been used in this station during last years, and it seems the lowest genetic diversity can be consistent with this issue in this breed. While, the rural livestock system of Zel and Lori-Bakhtiari is as the rams and ewes are housed and grazed together and there is no control over mating and inbreeding. Eydivandi et al. [45] reported the range of $H_o$ in Iranian domestic sheep breeds ranged from 0.343 up to 0.389. Al-Mamun et al. [16] reported almost the same $H_e$ in Australian sheep populations with 0.38, 0.31 and 0.34, in Merino (MER), Border Leicester (BL), and Poll Dorset (PD), respectively. Deniskova et al. [43] investigated genetic diversity of 25 Russian sheep breeds by the whole genome information and reported that Romanov breed had the lowest level of genetic diversity with an $H_e = 0.354$. Dávila et al. [46] suggested suitable $H_e$ greater than 0.5 for genetic diversity of a breed.

## Linkage disequilibrium and persistence of LD phase

Linkage disequilibrium was calculated separately for each breed using $r^2$. The average $r^2$ values between adjacent SNPs across autosomal chromosomes were different for each breed. The Baluchi breed had the highest level of LD and the Zel breed had the lowest level of LD across all distances. At distances up to 10Kb, the mean $r^2 \pm SD$ between adjacent SNPs were $0.388 \pm 0.324$, $0.353 \pm 0.311$, and $0.333 \pm 0.309$ for Baluchi, Lori-Bakhtiari and Zel, respectively. The average $r^2$ values presented a slight difference among autosomal chromosomes for each breed. In the analysis of LD, decay for distances from 0 to 50 Mb is shown in Fig 3, indicating the $r^2$ values decreased rapidly with increasing distances between markers.

The minimum average values of $r^2$ were obtained at a distance of 10 to 50 Mb in all the breeds. Previous studies suggested an $r^2$ higher than 0.3 for GWAS [47], while an LD of more than 0.2 is considered essential for estimating genomic breeding values with around 0.85 accuracies [5]. Considering that in the current study, the average $r^2$ for the threshold of 0.2 was reached at a distance of 27 Kb for Zel and Lori Bakhtiari breeds and 41 Kb for the Balochi breed, it seems, the SNP chips with a marker density higher than 90K in the Zel and Lori-Bakhtiari breeds and 60K in the Baluchi breed will be needed for GS studies. At distances up to 10 Kb, the percentage of pairs of markers with $r^2 > 0.3$ was 48.4% (Baluchi), 44.3% (Lori-Bakhtiari), and 41.1% (Zel). The differences in the LD pattern can be due to the selection process, population structure, $N_e$, marker allele frequencies, and the average of the distances between SNPs [48, 49]. In this study, LD is measured by $r^2$ because compared to $D'$, it is less influenced by sample size and allele frequency. Using $D'$ for small sample sizes can lead to overestimates in LD [50]. Khatkar et al. [6] suggested a minimal sample size of 75 for $r^2$, but Bohmanova et al. [49] showed a sample size of 22, tended to overestimation of $r^2$. In this regard, the

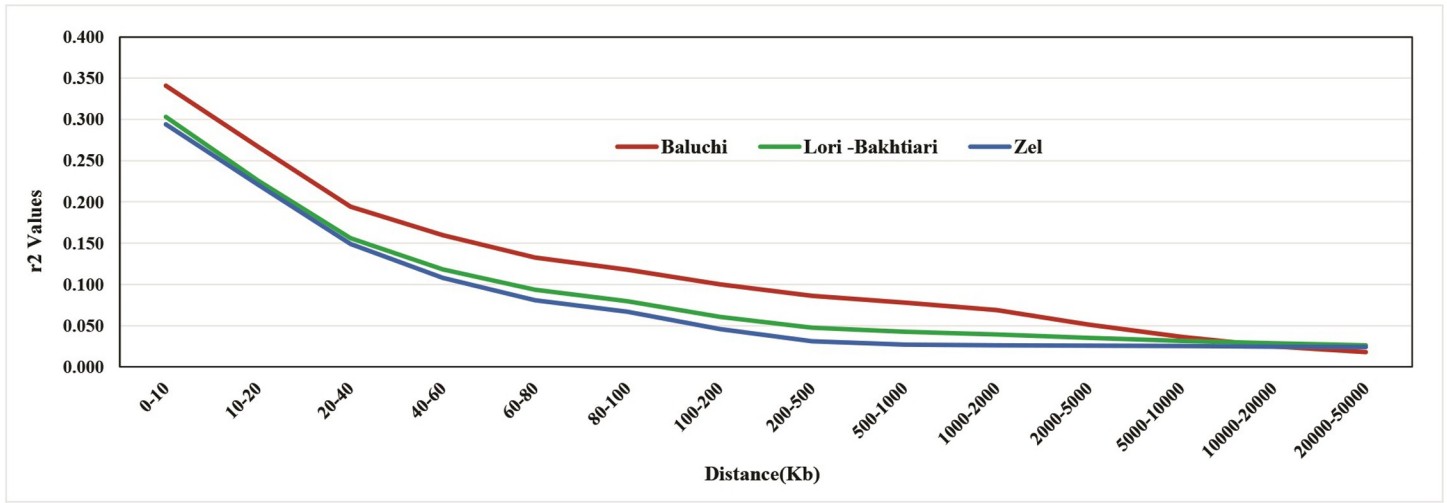

**Fig 3. The decay of mean r² as a function of physical distance for each breed.** The average $r^2$ values were estimated at distances up to 50 Mb. Red, green and blue colors are showing Baluchi, Lori-Bakhtiari, Zel sheep breeds in this Fig, respectively.

estimated $r^2$ values were corrected for sample sizes. Baluchi indicated higher levels of LD compared to other breeds, this was probably due to the upward selection intensity and the small sample size of breeding stations. Selection, over generations, on the allele of interest will increase the frequency of adjacent alleles known as "selective sweep" and lead to increased linkage disequilibrium and decreased diversity at these points, so the extent of linkage disequilibrium depends on the selection intensity in the breeding program [51].

Persistence of the LD phase is important for GS in multiple breeds and genome-wide association studies because it can be used to characterize the marker density and generate a multi-breed training population. Our results revealed that the highest persistence of the LD phase presented itself between Zel and Lori-Bakhtiari, followed by Zel and Baluchi and finally Lori-Bakhtiari and Baluchi (Fig 4). The high persistence of the LD phase between Zel and Lori-

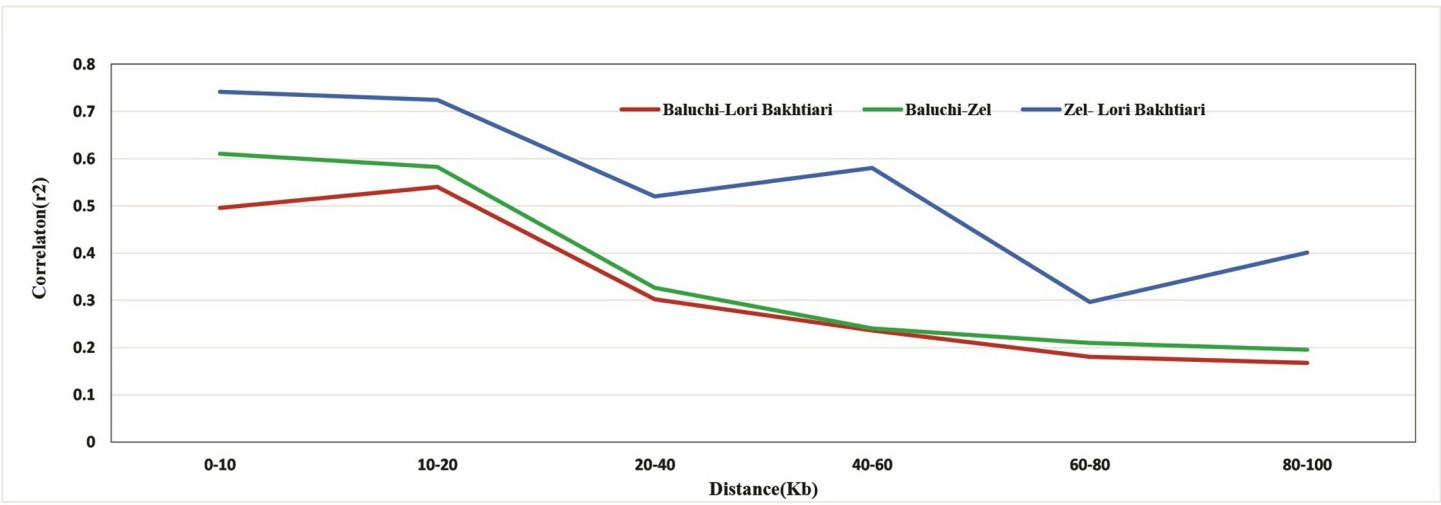

**Fig 4. Correlation of the persistence of LD phase between breeds in distance categories for SNP in 0–100 kb across the genome.** Blue, green and red colors are showing the persistency between Zel-LoriBakhtiari, Baluchi-Zel and Baluchi-LoriBakhtiari sheep breeds, respectively.

Bakhtiari represents a genetic closeness between these two breeds, which is consistent with the population structure results described above. The expectancy persistence of phase decreased with increasing distance between SNP. However, the persistence of phase decay showed rather erratic behavior between Lori-Bakhtiari and Zel with the exception at two points (from 40 to 60 and 80–100 Kb) and between Lori-Bakhtiari and Baluchi at one point (from 10 to 20 Kb) where there was an increase in the persistence of phase (Fig 4).

### Haplotype blocks structure

A summary of the analysis for the haplotype blocks structure is indicated in Table 3. Haplotype blocks are defined as long stretches of a chromosome that have low recombination rates [52].

Knowledge of the structure of haplotype blocks provides useful information for GS studies. The pattern of haplotype blocks is different on chromosomes due to a variety of factors such as heterogeneous recombination, population bottlenecks, density of markers, mating among populations with different allele frequencies, and selection intensity on the regions of the genome [53]. The distribution of haplotype blocks per autosomal chromosomes shows that the total length of blocks was 46086 Kb (1.90% spanning percentage of the genome), 14871 Kb (0.61%), and 14633 Kb (0.60%) for Baluchi, Zel, and Lori-Bakhtiari, respectively (Table 3). In this study, we found 1446 haplotype blocks in Baluchi, 604 in Zel, and 636 in Lori-Bakhtiari (consisting of 2 or more SNPs). The coverage percentage of SNPs on autosomal chromosomes was 8.194%, 3.195%, and 3.371% in Baluchi, Zel, and Lori-Bakhtiari, respectively. Baluchi seems to have experienced more intense selection than other breeds. Chromosome 2 showed the largest number of blocks and the total block length among all breeds (Table 3). The coverage of markers on chromosome 2 was higher than other chromosomes. Al-Mamun et al. [16] reported the longest block was detected on OAR10 for Merino (MER), Poll Dorset (PD), and two crossbred populations ($F_1$ crosses of Merino and Border Leicester (M×B) and M×B crossed to Poll Dorset (M×B×P)).

### Effective population size ($N_e$)

The $N_e$ was estimated for all three breeds based on Eq 7 and Eq 8. The results were similar with both methods and, showed the reduction of the $N_e$ by increasing the share of haplotypes. Note

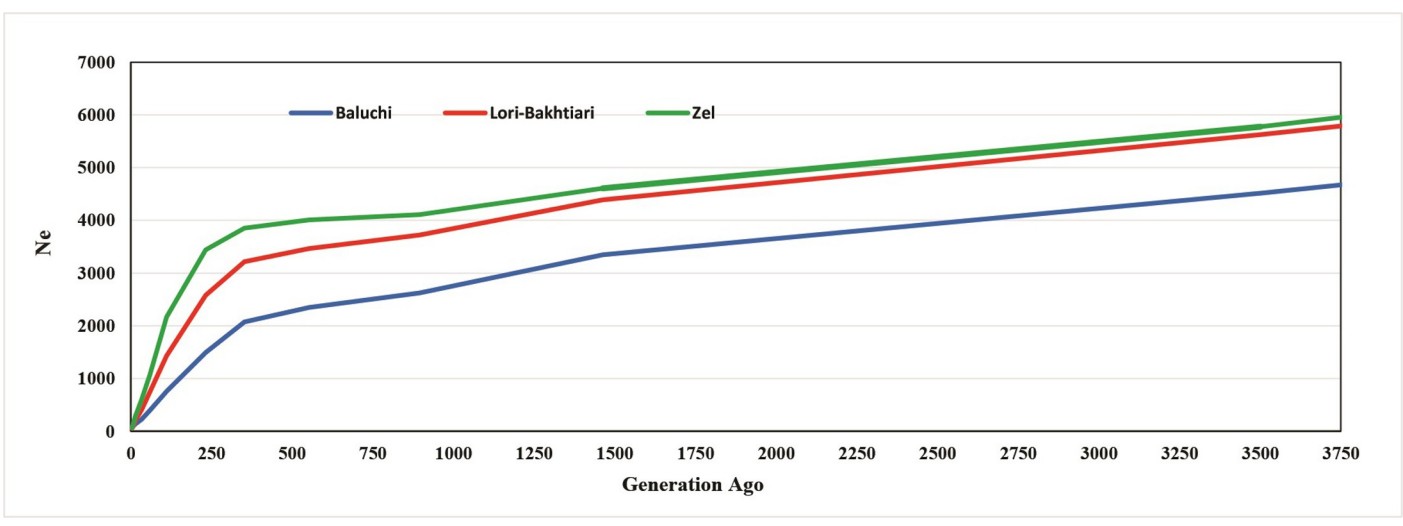

**Fig 5. Trend of effective population size ($N_e$) in Iranian sheep breeds across past generations.**

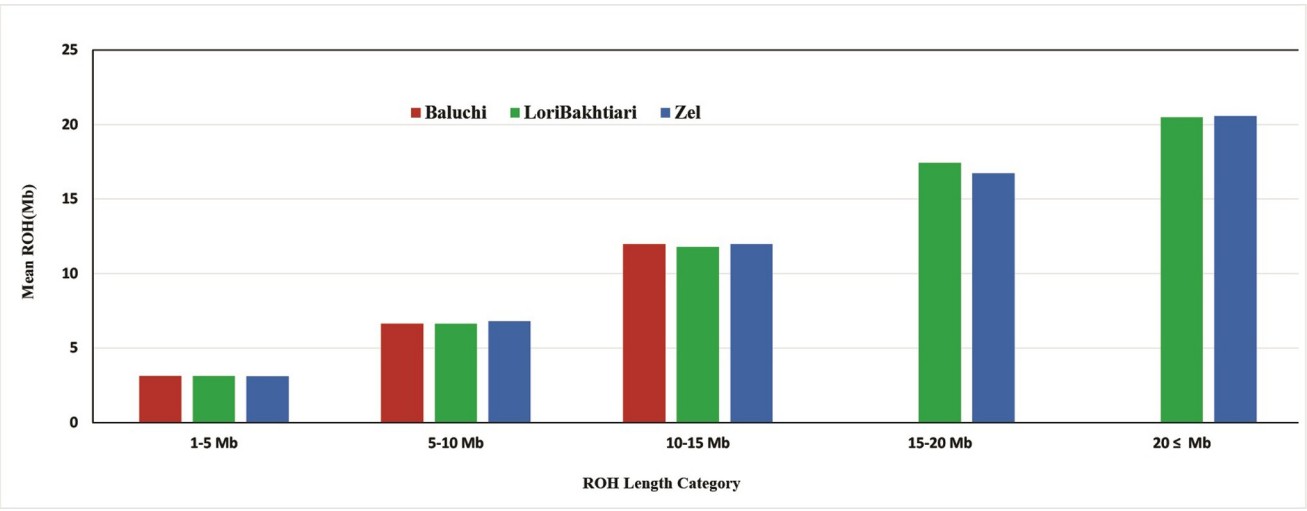

**Fig 6. The average of ROH and ROH length categories in the Baluchi, Zel and Lori-Bakhtiari sheep breeds.** The Baluchi ROH length is in the first three categories, between 1–15 Mb, and is not long enough to be in the >15 Mb categories. Red, green and blue colors are showing the individuals of Balochi, Lori-Bakhtiari and Zel sheep breeds, respectively.

that both of the methods use the $r^2$ values combined with marker distances and recombination rate to estimate $N_e$, but SNeP software estimates $r^2$ and then consider adjusted $r^2$ with recombination rate and corrects for the occurrence of mutations. In this study, estimates of $N_e$ showed a downward slope (Fig 5). The slope of Baluchi was stronger than the Zel and Lori Bakhtiari.

Evolutionary processes such as the rates of genetic drift, loss of genetic variability, the effectiveness of selection, and gene flow depend on $N_e$ [54]. Kimura and Ohta [55] showed the time required for the fixation of one allele depends on $N_e$ and allele frequency. According to the process of reducing $N_e$ obtained in the present study (Fig 5), and considering the results reported by Moradi et al. [15], it seems Zel and Lori Bakhtiari were diverged and the breeds

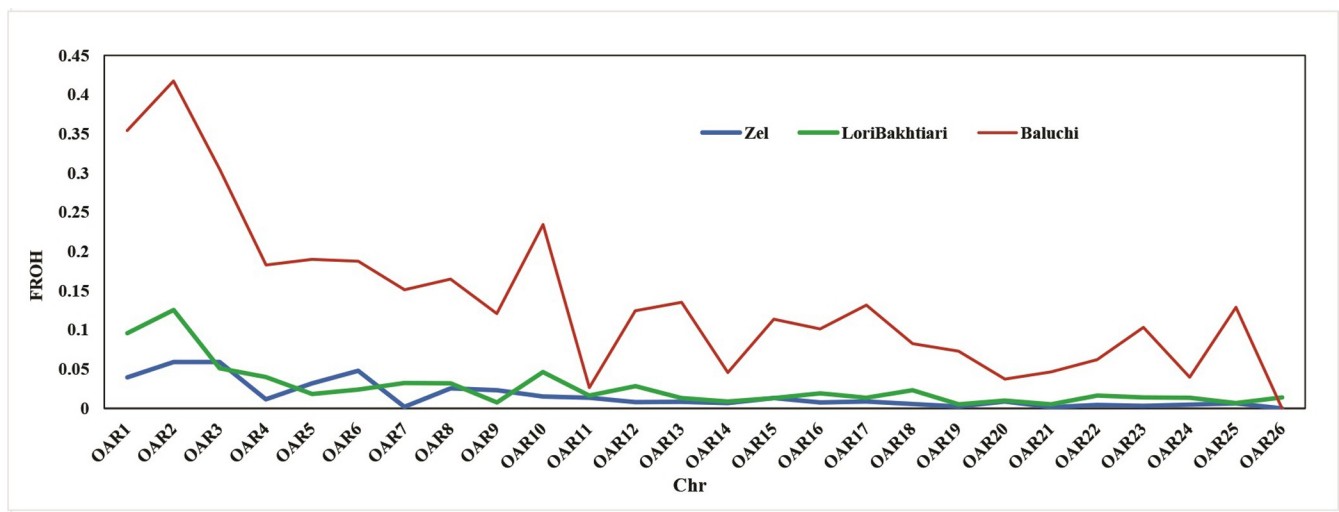

**Fig 7. The inbreeding coefficient ($F_{ROH}$) derived from ROH in autosomal chromosomes.** The $F_{ROH}$ in Baluchi for all autosomal chromosomes was higher than other breeds and the highest was in chr 2. The FROH for Baluchi, Lori-Bakhtiari and Zel is exhibited in red, green and blue lines respectively.

**Table 3. Haplotype block summary and statistics for the SNPs that passed quality control (QC): The percentage of chromosome covered by blocks, total block length and percentage of SNPs in blocks on a per chromosome.**

| Chr | Number of blocks | | | % of Chr length in blocks | | | Block coverage length (Kb) | | | % of SNPs in blocks | | |
|---|---|---|---|---|---|---|---|---|---|---|---|---|
| | Zel | Lori-Bakhtiari | Baluchi | Zel | Lori-Bakhtiari | Baluchi | Zel | Lori-Bakhtiari | Baluchi | Zel | Lori-Bakhtiari | Baluchi |
| 1 | 67 | 71 | 181 | 0.40 | 0.55 | 1.61 | 1112 | 1524 | 4423 | 3.07 | 3.35 | 8.56 |
| 2 | 87 | 90 | 181 | 1.47 | 1.23 | 2.94 | 3661 | 3068 | 7319 | 4.75 | 4.91 | 10.27 |
| 3 | 72 | 63 | 140 | 0.78 | 0.59 | 2.08 | 1757 | 1315 | 4666 | 3.98 | 3.52 | 8.51 |
| 4 | 29 | 27 | 68 | 0.44 | 0.57 | 1.81 | 529 | 685 | 2155 | 2.84 | 2.88 | 7.25 |
| 5 | 31 | 30 | 72 | 0.42 | 0.61 | 2.56 | 445 | 656 | 2740 | 3.37 | 3.56 | 9.50 |
| 6 | 24 | 32 | 73 | 0.53 | 0.66 | 2.58 | 620 | 766 | 3011 | 2.56 | 3.43 | 10.10 |
| 7 | 25 | 29 | 61 | 0.32 | 0.38 | 1.34 | 316 | 381 | 1336 | 2.82 | 3.26 | 7.50 |
| 8 | 24 | 29 | 60 | 0.28 | 0.48 | 1.49 | 257 | 436 | 1351 | 2.82 | 3.59 | 7.84 |
| 9 | 31 | 39 | 76 | 0.8 | 1.06 | 2.66 | 754 | 998 | 2510 | 4.01 | 5.06 | 10.96 |
| 10 | 26 | 24 | 51 | 2.25 | 1.60 | 2.07 | 1911 | 1357 | 1757 | 4.93 | 4.11 | 9.50 |
| 11 | 12 | 11 | 28 | 0.39 | 0.22 | 1.04 | 239 | 138 | 635 | 2.68 | 2.44 | 6.35 |
| 12 | 21 | 18 | 47 | 0.76 | 0.25 | 3.37 | 597 | 195 | 2652 | 3.06 | 2.64 | 8.55 |
| 13 | 18 | 23 | 47 | 0.35 | 0.83 | 2.94 | 295 | 688 | 2440 | 2.75 | 3.77 | 8.85 |
| 14 | 11 | 13 | 26 | 0.22 | 0.20 | 1.22 | 136 | 126 | 763 | 2.38 | 2.75 | 6.84 |
| 15 | 14 | 18 | 43 | 0.20 | 0.25 | 2.24 | 158 | 198 | 1794 | 2.16 | 2.76 | 8.03 |
| 16 | 11 | 11 | 29 | 0.21 | 0.18 | 1.44 | 153 | 126 | 1030 | 1.79 | 1.80 | 5.49 |
| 17 | 17 | 12 | 35 | 1.00 | 0.62 | 1.07 | 719 | 451 | 771 | 3.82 | 2.84 | 6.64 |
| 18 | 13 | 18 | 39 | 0.16 | 0.45 | 0.76 | 108 | 301 | 512 | 2.20 | 3.26 | 6.65 |
| 19 | 13 | 12 | 30 | 0.34 | 0.22 | 1.56 | 205 | 135 | 940 | 3.77 | 2.45 | 7.00 |
| 20 | 7 | 8 | 19 | 0.17 | 0.60 | 0.68 | 85 | 305 | 337 | 1.66 | 2.22 | 4.50 |
| 21 | 8 | 7 | 18 | 0.17 | 0.13 | 0.32 | 87 | 67 | 162 | 2.40 | 2.13 | 4.94 |
| 22 | 12 | 15 | 32 | 0.35 | 0.26 | 1.62 | 178 | 132 | 817 | 2.95 | 3.30 | 8.03 |
| 23 | 8 | 13 | 22 | 0.14 | 0.22 | 0.64 | 86 | 139 | 399 | 1.75 | 2.91 | 5.31 |
| 24 | 9 | 8 | 19 | 0.18 | 0.17 | 0.49 | 77 | 70 | 205 | 2.93 | 2.61 | 6.20 |
| 24 | 8 | 10 | 31 | 0.69 | 0.68 | 2.30 | 314 | 311 | 1039 | 2.50 | 2.84 | 9.15 |
| 26 | 6 | 5 | 18 | 0.16 | 0.15 | 0.73 | 72 | 65 | 322 | 1.60 | 1.36 | 5.16 |
| **Total** | **604** | **636** | **1446** | **0.61** | **0.60** | **1.89** | **14871** | **14633** | **46086** | **3.195** | **3.371** | **8.194** |

separated from each other, around 1100–1300 generations ago (~5000–6000 years ago with considering ~4.5 generation interval).

The estimated $N_e$ at four generations ago for Baluchi, Lori-Bakhtiari, and Zel were 57, 65, and 66, respectively, which are in range of 50 to 100, the $N_e$ range recommended for conservation [56]. Mastrangelo et al. [57] estimated contemporary $N_e$ equal 25 in Barbaresca sheep breed in southern Italy. The reduction of $N_e$ is probably due to an increase in inbreeding rate and a reduced genetic diversity by domestication, breed formation, and artificial breeding technologies [29]. The main reason for the reduction of $N_e$ in the recent generations in Iranian breeds, observed in the current study, would be the low efficiency of production. So, the use of well-designed breeding programs is necessary to control the loss of genetic diversity, increased rate of inbreeding and reduce the risk of extinction.

## Inbreeding coefficient

The inbreeding coefficients estimated by four different methods including $F_{GRM}$, $F_{HOM}$, $F_{UNI}$, and $F_{ROH}$ are presented in Table 4.

Due to incomplete and careless recorded pedigree for the studied sheep breeds, the use of the genome-wide data for the estimation of inbreeding, and assessing their accuracy is important. The main advantage of genomic information is to realize true proportion of genome-

**Table 4. Estimates of inbreeding using different methods for Iranian sheep breeds.**

| Methods[*] | Baluchi | Zel | Lori-Bakhtiari |
|---|---|---|---|
| $F_{GRM}$ | -0.0179 | -0.0025 | -0.0157 |
| $F_{HOM}$ | -0.0178 | -0.0024 | -0.0156 |
| $F_{UNI}$ | -0.0178 | -0.0024 | -0.0156 |
| $F_{ROH}$ | 0.0410 | 0.0090 | 0.0153 |

[*]Inbreeding coefficients calculated from the genomic relationship matrix, the excess of homozygosity, the correlation between uniting gametes, and derived from runs of homozygosity were showed by $F_{GRM}$, $F_{HOM}$, $F_{UNI,}$ and $F_{ROH}$ respectively.

wide relationship between two individuals and inbreeding for specific regions of the genome [58]. While the pedigree-based inbreeding assumes an equal chance for the two alleles at the same locus on two homologous chromosomes [59], in many loci the two alleles may have different chances for selection, as a result the assumption will be true only under an infinitesimal model [60].

The estimates from the first three methods were almost similar in all breeds. The results revealed that the average of inbreeding calculated by the genomic relationship matrix ($F_{GRM}$), the excess of homozygosity ($F_{HOM}$) and the correlation between uniting gametes ($F_{UNI}$) are higher in Zel than other two breeds. The ROH is defined as the lengths of homozygous genotypes above 1Mb that contain only up to one heterozygous genotype [61]. $F_{ROH}$ is the inbreeding coefficient derived from ROH based on molecular approaches that allow for recombination and mutation to apprehend relatedness among founders [62]. Estimation of inbreeding by $F_{GRM}$, $F_{HOM}$, and $F_{UNI}$ methods are sensitive to allelic frequencies and the number of copies of reference alleles for $i^{th}$ SNP [63]. Also, these methods cannot distinguish alleles that are IBD or IBS [64]. The use of ROH leads to accurately estimated levels of autozygosity among individuals because this method has better accuracy in distinguishing between IBD and IBS [65]. Zanella et al. [65] studied a comparison of different methods for estimating inbreeding values using genomic ($F_{ROH}$) and pedigree data in commercial pigs indicated that the use of $F_{ROH}$ has been more reliable. Their study showed that the estimation of inbreeding using the SNP-by-SNP method overestimates the levels of inbreeding compared to $F_{ROH}$ because it uses the frequency of homozygous genotypes, including both IBD and IBS alleles. Comparison of ROH and other methods, derived from GCTA software, demonstrate that ROH has the lowest sensitivity to allelic frequencies, and this method is very capable of distinguishing between IBD and IBS. In this study, the lowest $F_{ROH}$ was observed in Zel breed and this is in constant with the results obtained for $N_e$, as described previously, the largest $N_e$ was observed in Zel than other breeds in this study.

While the previous methods display the inbreeding coefficient for recent generations and does not provide any information about population history, long ROH indicates recent inbreeding that may be due to the mating of close relatives and the decrease of $N_e$, but shorter ROH suggests the reduction of genetic diversity, bottlenecks, and founder effects in the initial population [16, 66]. The measurement of ROH in the studied breeds indicates that Zel and Lori-Bakhtiari have the longest ROH containing lengths higher than 20Mb, which may be due to selection intensity, reduced effective size of the population, and increased inbreeding in recent generations (Fig 6). Among the breeds, Baluchi had the shortest ROH which was probably due to inbreeding in ancestral generations and a small ancestral population.

Chromosomal inbreeding coefficients are shown in Fig 7. Among breeds, Baluchi had the maximum sum chromosomal run of homozygosity and inbreeding coefficient for all

chromosomes. The highest contiguous homozygous stretches in the Baluchi and Lori-Bakhtiari breeds were for OAR 2, 1, 3 and 10. The highest chromosomal inbreeding coefficients in Zel were for OAR 2 and 3. This may be caused by lower recombination, followed by increasing homozygosity and inbreeding coefficient.

## Conclusion

This study provides a comprehensive assessment of genetic structure, linkage disequilibrium (LD) and several other genetic diversity parameters, including gene diversity ($H_e$), $N_e$ and genomic inbreeding coefficients in Iranian sheep breeds. The PCA and admixture analysis displayed a clear genetic differentiation of the breeds. Genome-wide study of LD in Iranian sheep breeds showed that the Baluchi breed has higher levels of LD and haplotype blocks than other breeds, which is agreed with the results of genetic diversity, $N_e$ and inbreeding coefficient analysis in this breed. We found that the amount of LD was relatively small between the adjacent SNPs and it decreased rapidly by increasing the distance between the markers in Baluchi breed. Also, the persistence of the LD phase presented the highest compatibility between Zel and Lori-Bakhtiari, which is consistent with the existence of a common ancestor in the past. This results provide insights into the influence of selection within these breeds and provide useful knowledge that will contribute to design appropriate and successful genomic selection and conservation programs. Take note that this results can be also useful for constructing a multi-breed training population and SNP array designing, however, further investigation with high marker density and more animals, are required to confirm our results.

## Acknowledgments

The authors gratefully acknowledge the Animal Breeding Center of Iran (ABCI) for access to the records and animals of the Iranian breeds. Thanks to the staff of the University of Tehran, Animal Science Research Institute of Iran and AgResearch, who helped and supported this research. The authors also acknowledge the financial contributions of Animal Science Research Institute of Iran, Mobarakandish Institute and AgResearch, New Zealand.

## Author Contributions

**Conceptualization:** A. Nejati-Javaremi, M. H. Moradi.

**Data curation:** M. Moradi-Sharbabak, M. Gholizadeh.

**Formal analysis:** S. Barani, M. H. Moradi.

**Funding acquisition:** A. Nejati-Javaremi, M. Moradi-Sharbabak.

**Investigation:** S. Barani.

**Methodology:** M. H. Moradi.

**Project administration:** A. Nejati-Javaremi.

**Resources:** M. H. Moradi.

**Software:** H. Esfandyari.

**Writing – original draft:** S. Barani, M. H. Moradi, H. Esfandyari.

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
