## [Decision Letter · Decision Letter 0]

1 Mar 2023

PONE-D-22-33894Genome-wide study of linkage disequilibrium, population structure, and inbreeding in Iranian indigenous sheep breedsPLOS ONE

Dear Dr. Moradi,

Thank you for submitting your manuscript to PLOS ONE. After careful consideration, we feel that it has merit but does not fully meet PLOS ONE’s publication criteria as it currently stands. Therefore, we invite you to submit a revised version of the manuscript that addresses the points raised during the review process.

Based on the advice received, I feel that your manuscript could be reconsidered for publication if you're willing to make Substantial modifications. I ask that in preparing your revised manuscript you consider all comments carefully. 

Please be aware that the revision required is really substantial, and you should also keep in mind that I may ask the reviewers for their second opinion, when the revised version of your manuscript reaches me. It is also of great importance that the quality of writing is considerably Improved. This is of critical importance.

We look forward to receiving your revised manuscript.

Kind regards,

Shamik Polley, Ph.D

Academic Editor

PLOS ONE

Journal Requirements:

"No,"

"NO authors have competing interests"

6. Please upload a new copy of all figures as the detail is not clear. Please follow the link for more information: " ext-link-type="uri" xlink:type="simple">https://blogs.plos.org/plos/2019/06/looking-good-tips-for-creating-your-plos-figures-graphics/"
https://blogs.plos.org/plos/2019/06/looking-good-tips-for-creating-your-plos-figures-graphics/

Reviewers' comments:

Reviewer's Responses to Questions

**Comments to the Author**

1. Is the manuscript technically sound, and do the data support the conclusions?

Reviewer #1: Yes

Reviewer #2: Yes

2. Has the statistical analysis been performed appropriately and rigorously? 

Reviewer #1: Yes

Reviewer #2: Yes

3. Have the authors made all data underlying the findings in their manuscript fully available?

Reviewer #1: Yes

Reviewer #2: Yes

4. Is the manuscript presented in an intelligible fashion and written in standard English?

Reviewer #1: Yes

Reviewer #2: Yes

5. Review Comments to the Author

Reviewer #1: The manuscript entitled “Genome-wide study of linkage disequilibrium, population structure, and inbreeding in Iranian indigenous sheep breeds” is interesting. In the study, the authors used OvineSNP50 Bead Chip array to estimate and compare LD, genetic diversity, effective population size and genomic inbreeding of three Iranian indigenous sheep breeds. The authors found that the Baluchi breed has higher levels of LD and haplotype blocks than other two breeds. Estimation of genomic inbreeding showed the highest inbreeding coefficient in Baluchi and the lowest in Zel breed that was explained by high pressure of artificial selection on Baluchi breed. Overall, the findings are interesting and the manuscript is well structured and well written. My only concern is that the sample size is less specially for Lori-Bakhtiari (=47) and Zel (=47) breeds. Otherwise, the manuscript has merit and may be accepted for publication.

Reviewer #2: The article entitled “Genome-wide study of linkage disequilibrium, population structure, and inbreeding in Iranian indigenous sheep breeds ” focuses on the knowledge of linkage disequilibrium, genetic structure and genetic diversity in three Iranian local sheep breeds. This study is of wide interest to the community, but I have following concerns which needs to be addressed.

1.In the introduction section, more breed information for Baluchi , Lori-Bakhtiari and Zel is needed.

2. why you state more information for QTL, GS, and LD in the introduction section?please explain it. I personally feel that the current organizational structure of the introduction section is unreasonable, and more valuable information should be provided by the author.

3.Line 92, please the name of database,or provide the accession number.

4. line 113, how many SNPs was remain? Please provide it in the text.

5. line 118, provide the author name

6.Line200-202, please rewrite this sentence.

7.Line 914, please fixed the subtitle.

8.Please write the results and discussion, respectively. The current write way is difficult for readers to clearly understand the author's intention.

9.Improve discussion and conclusion. More utility from this work should be signified.

6. PLOS authors have the option to publish the peer review history of their article (what does this mean?). If published, this will include your full peer review and any attached files.

Reviewer #1: **Yes: **Arun Kumar De

Reviewer #2: No

---

## [Author Response · Author response to Decision Letter 0]

13 Apr 2023

We have provided a systemic response to each question/comment and alterations on a point by point basis as attached in the " Responses to Reviewers Comments" file and appended below at the end of this file.

---

## [Editor Report · Decision Letter 1]

27 Apr 2023

PONE-D-22-33894R1Genome-wide study of linkage disequilibrium, population structure, and inbreeding in Iranian indigenous sheep breedsPLOS ONE

Dear Dr. Moradi,

Thank you for submitting your manuscript to PLOS ONE. After careful consideration, we feel that it has merit but does not fully meet PLOS ONE’s publication criteria as it currently stands. Therefore, we invite you to submit a revised version of the manuscript that addresses the points raised during the review process.

We look forward to receiving your revised manuscript.

Kind regards,

Shamik Polley, Ph.D

Academic Editor

PLOS ONE

Journal Requirements:

Additional Editor Comments:

I appreciate the authors' corrections. Still not satisfied with the article's figures' resolution. There are a few minor corrections listed below.

Line 249: (Fig 1Error! Reference source not found.)          

-        Please check it and modify it accordingly.

Line 301: Fig 1.

-      It seems to fit in fig 3. Please check it again and correct it in the manuscript.

---

## [Author Response · Author response to Decision Letter 1]

6 May 2023

Additional Editor Comments:

I appreciate the authors' corrections. Still not satisfied with the article's figures' resolution. 

- All the figures were replaced with the higher resolutions ones (in both pdf and tif formats) to address the Editor's concern.

There are a few minor corrections listed below.

Line 249: (Fig 1Error! Reference source not found.)

- Please check it and modify it accordingly.

- It was modified as the Editor mentioned.

Line 301: Fig 1.

- It seems to fit in fig 3. Please check it again and correct it in the manuscript.

- While the authors thank you very much for your deeply thoroughness, all the figures were checked in the main text again and the mistakes were corrected.

---

## [Editor Report · Decision Letter 2]

17 May 2023

Genome-wide study of linkage disequilibrium, population structure, and inbreeding in Iranian indigenous sheep breeds

PONE-D-22-33894R2

Dear Dr. Moradi,

We’re pleased to inform you that your manuscript has been judged scientifically suitable for publication and will be formally accepted for publication once it meets all outstanding technical requirements.

Kind regards,

Shamik Polley, Ph.D

Academic Editor

PLOS ONE
---

## [Editor Report · Acceptance letter]

22 May 2023

PONE-D-22-33894R2 

Genome-wide study of linkage disequilibrium, population structure, and inbreeding in Iranian indigenous sheep breeds 

Dear Dr. Moradi:

I'm pleased to inform you that your manuscript has been deemed suitable for publication in PLOS ONE. Congratulations! Your manuscript is now with our production department. 

Kind regards, 

on behalf of

Dr. Shamik Polley 

Academic Editor

PLOS ONE